# Raising Suicide in Medical Appointments—Barriers and Facilitators Experienced by Young Adults and GPs: A Mixed-Methods Systematic Review

**DOI:** 10.3390/ijerph20010822

**Published:** 2023-01-01

**Authors:** Debra Osborne, Kathleen De Boer, Denny Meyer, Maja Nedeljkovic

**Affiliations:** Centre for Mental Health, Swinburne University of Technology, P.O. Box 218, Hawthorn, VIC 3122, Australia

**Keywords:** young adults, suicide, self-harm, primary care, primary care physicians, general practitioners

## Abstract

The aim of this review was to understand the barriers and facilitators facing GPs and young adults in raising and addressing suicide in medical appointments. A mixed-methods systematic review was conducted of qualitative and quantitative studies. The focus was papers that explored barriers and facilitators experienced by young adults aged 18 to 26, and GPs working in primary care environments. Nine studies met the inclusion criteria. Four studies provided information on young adults’ views, four on GPs, and one considered both GP and young adults’ viewpoints. Nine barrier and seven facilitator themes were identified. Unique to this review was the recognition that young adults want GPs to initiate the conversation about suicide. They see this as a GP’s responsibility. This review further confirmed that GPs lack the confidence and skills to assess suicide risk in young adults. Both findings combined could explain previous results for reduced identification of suicide risk in this cohort. GP training needs considerable focus on addressing skill deficiencies and improving GP confidence to assess suicide risk. However, introducing suicide risk screening in primary care for young adults should be a priority as this will overcome the need for young adults to voluntarily disclose thoughts of suicide.

## 1. Introduction

Extensive work is being undertaken worldwide to reduce rates of suicide. Yet, over the past few decades, in high-income countries such as Australia, Canada, the UK, and USA, suicide rates in young adults have been increasing [1]. General practitioners (GPs) present an opportunity for early detection and intervention. In Australia, most young adults attend a GP regularly [2], averaging at least three visits per year with females attending more frequently than males (4.3 vs. 3.2) [3]. Prior to suicide, adults contact GPs more than any other health professional with higher rates of contact compared to age-matched peers [4,5], with young adults showing similar patterns [6,7]. It has been estimated that in the year and month prior to suicide, 71% and 33% of young adults, respectively, will present to a GP for help [8]. However, it is not clear whether young adults’ thoughts of suicide are directly discussed and assessed within these GP consultations.

The majority of young adults who are experiencing thoughts of suicide or suicidal behaviours do not raise these thoughts when they attend a GP appointment [9,10]. Non-disclosure of suicidal thoughts is common amongst all age groups. One study found that 40% of all adults did not disclose their thoughts of suicide to anyone [11]. Furthermore, younger adults (aged 18–39) have been found to be half as likely as older adults (over 39) to disclose thoughts of suicide at the last GP consultation prior to a suicide attempt (11.3% vs. 21.9%) [12]. Thus, it would appear that young adults, despite an increased frequency in attending the GP when experiencing thoughts of suicide, do not necessarily raise these thoughts or concerns with GPs. 

Observational and experimental research has found that GPs do not always actively explore suicide risk in a medical appointment even when a diagnosis such as depression would suggest that this is necessary [13,14,15,16]. Studies, mainly with adults presenting with depression, have found that only 36% to 66% of GPs explore suicide risk [13,14,15]. However, GPs were found to be more likely to explore risk when a patient’s depression presentation was judged as severe [14,15], was a new episode [13], or they also presented with psychosocial stress rather than a physical complaint [14]. Interestingly, GPs were also more likely to explore risk when prompted by a patient enquiry; for example, when a patient requests medication [15]. Similar findings have been made in other studies exploring the frequency of mental illness identification in young adults. Young adults who perceived themselves as having a mental illness were six times more likely to be identified as having a probable mental illness by the GP [16]. Overall, these studies suggest that GPs are responding to patients’ concerns rather than proactively assessing risk and are also relying upon young adults to raise concerns that need to be explored and discussed in a medical appointment.

Studies have sought to understand the barriers and difficulties GPs experience in addressing suicide risk in patient consultations across all age groups. Barriers identified include: insufficient time to conduct suicide risk assessments in appointments [17,18,19,20]; lack of tools to use to conduct the risk assessments [19]; low confidence in their skills and lack of training [17,18]. Furthermore, conversation analysis of patient consultations has found that when doctors (GPs and psychiatrists) do ask about self-harm and suicide, many phrase the question in such a way that patients are more likely to answer in the negative [21,22]. Even when the patient responses were ambiguous, doctors often do not explore further, thus missing the opportunity to thoroughly assess risk and also to provide validation and support to a patient who may be distressed [21]. Interventions such as training for GPs have been implemented to address some of these concerns, but to date, these have mainly been focused on training GPs to manage adults with depression with limited focus on young adults and suicide [23].

Considering that GPs are often the first professional help accessed by most young adults at risk of suicide, it is important to understand what prevents young adults from raising and discussing suicidal thoughts and behaviours, such as self-harm, planning, and suicide attempts, with their GP, and similarly what prevents GPs from proactively exploring suicide risk with these young adults. Given this, the current study aims to undertake mixed-methods thematic analyses to explore: (1) the barriers and facilitators facing GPs in detecting suicide risk in young adults in medical appointments; (2) the barriers and facilitators young adults experience in alerting GPs to the presence of suicidal thoughts and behaviours (STBs); (3) the possible dynamics within the relationship or medical appointment that prevent STBs and suicide risk being voiced, detected, and addressed.

## 2. Method

A protocol for this review was developed and lodged with PROSPERO (International Prospective Register of Systematic Reviews) on the 30th August 2021 (CRD42021275641). Reporting for this mixed-methods review is in accordance with the Preferred Reporting Items for Systematic Reviews (PRISMA) statement recommendations [24].

### 2.1. Search Strategy

Systematic searches were conducted in four academic databases: Scopus, Web of Science, PsychINFO, and CINAL. Scopus and Web of Science were used as the prime search databases as they were expected to hold the majority of the research related to GPs in primary care. PsychINFO and CINAL were used to identify any other research with young adults published outside medical journals. The search subjects used included GPs or “general practitioners” or “primary care physicians” AND “suicide (expressed as suic*) or “self-harm” or “non-suicidal self-injury” or Non-Suicidal Self-Injury (NSSI) AND “young adult” or “young adults”. Suicide was broadened to include self-harm including NSSI as both terms are often used to denote suicide attempts and deliberate self-harm regardless of intent [25,26,27]. Reference lists and citations of included studies were reviewed to identify additional relevant papers but none were identified. In addition, reference lists of recent systematic reviews focusing on young adults’ attitudes regarding primary care for mental health [28] and healthcare professionals’ attitudes [29] were also reviewed. This review identified an extra study [30]. For further information on database search results, refer to Appendix A provided as part of the Appendix A. 

### 2.2. Eligibility Criteria 

Studies were considered eligible if: (1) they were published before the date of the last search (12 June 2022); (2) the study included original data either in a qualitative or quantitative format on barriers and facilitators faced by either or both GPs and young adults in raising and addressing suicide and/or self-harm in primary care appointments; (3) they were in English; (4) participants were young adults or the majority of the sample were young adults aged between 18 and 26 and/or were GPs working in primary care settings. Studies that were not included in this review were: (1) those that included only adolescents (aged under 18) or where the majority of adults were aged over 26; (2) systematic reviews or non-peer-reviewed papers; (3) those focused on the barriers and facilitators experienced by young adults as part of the decision-making or help-seeking process prior to attending or deciding to attend a GP appointment; (4) those focused on GP difficulties in identifying and treating mental health issues in young adults; (5) those quantifying the prevalence of non-disclosure by young adults of suicidal thoughts and behaviours to GPs.

### 2.3. Study Selection

The database search identified 310 papers after duplicates were removed. The primary author (DO) conducted all the searches including hand searches and removed duplicate records. All identified papers were reviewed and selected for inclusion in two steps. In step one, the first (DO) and second author (KdeB) independently reviewed the titles and abstracts against the inclusion criteria and noted reasons for exclusion. At this step, the list of reasons for exclusion were refined and agreed to ensure uniformity in classification. In the second step, the 41 papers identified for full text review were examined independently by both authors (DO, KdeB) against the agreed inclusion criteria with reasons for the exclusion noted. In both steps, any disagreements between authors (DO, KdeB) were resolved in discussion with the fourth author (MN). Overall, 32 studies did not meet the inclusion criteria and were excluded from further analysis. Nine studies remained. The flow of studies as per PRISMA guidelines is shown in Figure 1. 

### 2.4. Data Extraction

A customised data extraction form was developed based upon a combination of the Cochrane Consumers and Communication data extraction template (2016) for study characteristics and the appropriate qualitative or quantitative Joanna Briggs Institute (JBI) SUMARI tool to assess methodological validity [31,32]. Prior to extraction of data, the form was tested by both researchers (DO and KdeB) to ensure consistency in the approach. Both researchers (DO and KdeB) then independently extracted data and completed the quality assessment for each study. Any differences were resolved by consensus and when necessary in consultation with the fourth author (MN). 

### 2.5. Data Analyses 

The strategy for data synthesis was based on the approach used by Kantor et al. [33]. The first step was to identify themes in the qualitative information [34] for the two populations under review (GPs and young adults). Researchers (DO and KdeB) first independently searched for patterns of meaning amongst the extracted barriers and facilitators documented in the qualitative studies, and grouped the barriers and facilitators according to possible themes. These themes were discussed and agreed by the two researchers (DO and KdeB) with subsequent review by the fourth author (MN). In the second step, the most frequently reported or top barriers and facilitators in the quantitative studies were applied to the identified themes from the qualitative analyses. 

## 3. Results

### 3.1. Study Characteristics

Overall, nine studies conducted between 2001 and 2021 were included in the systematic review with the majority (eight studies) conducted in the last ten years. Five studies were from the UK [35,36,37,38,39], three from Australia [40,41,42], and one from Nicaragua [30]. All have free universal healthcare for their population with primary care practices and clinics providing the initial access point for obtaining care [43]. The UK and Australia have a higher number of GPs (5.8 and 3.8 per 1000) per capita than Nicaragua (1.7 per 1000) [44] and the UK and Australian populations have greater access to healthcare compared to those in Nicaragua [43]. Four studies focused on barriers and facilitators experienced by GPs [30,37,39,41]; four focused on the young adult’s perspective [36,38,40,42]; one study [35] explored both GP and young adult perspectives on barriers and facilitators in raising and discussing suicide in a GP appointment. Table 1 details the characteristics of each study included in the review. 

### 3.2. Quality Assessment 

Seven of the nine studies were considered to be of good quality (*n* = 7) and rated at 80% or above. Two studies rated below 80% [30,38]. Refer to Appendix A for further information on the quality assessments for all studies. Overall, as the purpose of the review was exploratory, the two studies that rated below 80% were included as they were deemed to be of acceptable quality and provided an important contribution to the current research landscape. 

### 3.3. Thematic Analysis

The initial thematic analysis of the qualitative data items (178 data items) resulted in nine barrier themes (GPs—four; young adults—five) and seven facilitator themes (GPs—four; young adults—three). A review of the quantitative data items (6 data items) indicated no additional thematic information and so these data items, as per the protocol, were included into existing themes. For further information on the 184 data items identified, refer to excel spreadsheet Appendix A.

### 3.4. Barriers and Facilitators–GPs

The thematic analysis for GP barriers and facilitators was drawn from five studies. Three were conducted in the UK [35,37,39] with one each in Australia [41] and Nicaragua [30]. The resultant analysis is summarised in Table 2.

#### 3.4.1. Barrier 1. GP Attitudes Impede Enquiry about Suicide with Young Adults 

Across all the studies, GPs reported many negative attitudes when asked about the difficulties in having to proactively explore and discuss suicidal thoughts and behaviours (STB) with young adults. This ranged from negative emotions, cognitions, and beliefs to avoidance and stigma. Negative emotions and cognitions identified by GPs included: “feelings of discomfort, worry and uncertainty” [41]; finding it difficult to meet patients who expressed a wish to die [30]; some GPs described difficultly maintaining compassion for suicidal patients [39]. Avoidance was commonly reported: wanting to avoid problematic patients [30]; being reluctant to initiate the conversation about self-harm as it can be awkward and difficult [41]; not “want[ing] to open a can of worms” [35]. Stigma beliefs expressed included: that it is very difficult to predict suicide, therefore making suicide unpreventable; the belief that those who are truly suicidal will not seek help from their GP; that those expressing thoughts of suicide were “attention seeking” [37]. In Australia, one study identified that some GPs believed managing suicidal patients was outside their remit as a GP, and they were being forced to operate as an emergency service [41]. However, what was unclear from all the studies was the extent to which these expressed attitudes resulted in GPs not exploring suicide risk with young adults. 

#### 3.4.2. Barrier 2: GPs find Appointments with Young Adults Uniquely Difficult 

Most GPs related that appointments with young adults present unique challenges and identified a variety of reasons for this. First, GPs identified communicating with young adults was harder compared to older adults [30,39,41], and that, “in reality, it took two, three or more sessions to get to know a [young adults] … problem” [30]. Second, GPs reported that they found it difficult to identify STB in young adults. This included difficulty distinguishing suicidal thoughts from teenage angst [41], somatic complaints [30]; or a “cry for help” [30]. GPs indicated it was particularly difficult when the young adults were reluctant to reveal the true extent of their distress [41]. The third area of difficulty reported was in managing and negotiating confidentiality in appointments. For example, how to manage confidentiality when parents attended appointments [30,37,41] and when to disclose risk to others [37]. 

#### 3.4.3. Barrier 3: GPs Have Insufficient Time and Asking about Suicide Adds Stress to Their Day

Most GPs identified insufficient time in current appointments (typically 10 minutes in the UK and 15 minutes in Australia) as a barrier to raising and comprehensively exploring STB in young adults [30,35,37,41]. GPs acknowledged that young adults need time to develop sufficient trust in order to disclose STB [30,35] and that they themselves also needed extra time to develop rapport with a young adult to facilitate disclosure [41]. Insufficient time combined with heavy workloads made the process of exploration and assessing risk in young adults stressful for GPs. They described risk assessments as potentially taking up a lot of time that they did not have [35] and that the outcomes “could result in more responsibility and time pressure for the GP if they have to manage the risk identified” [41]. Once again, the extent to which this stopped GPs from exploring risk in young adults was unclear from the studies.

#### 3.4.4. Barrier 4: GPs Feel Ill-Equipped to Assess and Manage Suicidal Young Adults 

Many GPs reported being ill-equipped to effectively assess and also to then manage young adults who were suicidal within primary care [37]. GPs reported lacking training and the skills necessary to identify STB in young adults. This included: lack of knowledge, for example, of youth suicide predictors and how to accurately identify risk in young adults [30,37]; lack of therapeutic skills, for example, how to build rapport with young adults in order for them to disclose STB [30,41], and how to communicate with young adults generally [37]. GPs also reported a lack of available tools and guidelines to help them assess risk and guide actions to manage suicidal patients [37,41], and even when guidelines did exist, GPs were not aware of these or did not use [30]. Once STB had been identified in young adults, GPs reported facing multiple barriers to being able to manage these patients within primary care. For example, there was often no or inadequate inhouse support to help GPs to manage suicidal young adults within primary care, and accessing specialist services for those with higher risk was complicated with “dysfunctional referral pathways”, “complex and wishy-washy service entry criteria”, and “gaps in the healthcare system for those young adults who present with greater than low levels of risk” [37,41]. This lack of resources and the effort required to manage suicidal young adults within primary care, as well as difficulty accessing external support for higher-risk young adults, may limit GPs’ willingness to discuss and explore suicide with young adults. Once again, the extent to which this does limit GP behaviour was unclear.

#### 3.4.5. Facilitator 1: Training and Resources to Improve GP Ability to Detect and Assess Suicidal Young Adults

Many GPs believed additional training and resources would facilitate their ability to detect and address suicide risks in medical appointments [30,35,37,41]. Specific training customised to address GP concerns regarding assessing young adults for STB risk was highlighted, such as how to: involve family members in consultations without jeopardising doctor–patient confidentiality; assess suicide risk in young adults; manage challenging consultations with young adults [30,37,41]. GPs thought the provision of decision-making tools, guides, and flowcharts were necessary to conduct risk assessments to detect STB in young adults and to identify referral pathways for patients so they could better manage young adults experiencing STB within primary care [30,37,41].

#### 3.4.6. Facilitator 2: More Time for Consultations and Regular Reviews 

In most studies, GPs recognised that they needed additional time with young adults in order to better gather an in-depth understanding of the young person and their experiences, and to facilitate rapport and STB disclosure. Suggestions for increasing time with young adults included: offering time to the young person even if the GP runs late for the next appointment [41], increasing appointment times for young adults [30], and offering follow-up appointments [37]. 

#### 3.4.7. Facilitator 3: Improving GP Therapeutic Skills 

Most GPs recognised the importance of improving GP therapeutic skills to facilitate young adults disclosing STB [35,37,41]. The need for GPs to have good communication skills to improve disclosure was emphasised, for example: demonstrating a nonjudgmental approach to communicating with young adults [41], and “using open communication skills and direct questioning about suicide as a way of facilitating disclosure” [37]. GPs in one study also thought that fostering a trusting and collaborative relationship was an essential component of GP therapeutic skills. GPs indicated that if young adults felt they were being listened to and they trusted the GP, they were more likely to disclose STB [41]. 

#### 3.4.8. Facilitator 4: Better Systems to Support GP Ability to Manage Suicidal Young Adults in Primary Care

GPs raised systemic issues as barriers to better management of young adults with STB in primary care. They suggested addressing these barriers through the provision of in-house multidisciplinary care teams and integrated public health systems, so they could better manage young adults who are suicidal in primary care [30,41]. While these resources may not directly facilitate young adults to disclose STB in a medical appointment, it may be that, for GPs, knowing that they have the resources and support to manage young adults who are suicidal in primary care will, in turn, encourage or support them to proactively increase their enquiry of STB with young adults. 

### 3.5. Barriers and Facilitators–Young Adults

The thematic analysis of barriers and facilitators for young adults was drawn from five studies. Three were conducted in the UK [35,36,38] and two were in Australia [40,42]. The resultant analysis is summarised in Table 3. 

#### 3.5.1. Barrier 1: GP Responses (Verbal and Non-Verbal) Affect Decisions Whether to Disclose 

Young adults are attentive to GP verbal and non-verbal responses and they use this information to decide whether they can safely disclose thoughts of suicide now, and in the future. For example: in focus groups in one study, young adults reported that “interactions with GPs that convened an indifferent or impersonal attitude was seen as a barrier to honesty and disclosure” by the young person. They also reported perceived negative judgement from the GP and “conducting … [an] assessment in a ‘tick-box’ or formulaic way hindered disclosure” [40]. Two studies reported that young adults who experienced barriers at a GP consultation were less likely to open up in the future and also less likely to seek help [36,38]. Dismissal of reasons for self-harm or being referred to the nurse [35] or “GPs making unhelpful comments”, including assumptions made about the young person’s situation [36], led to young adults feeling their concerns were minimised or trivialised. For some young adults, not being heard within a GP appointment further increased their hopelessness and self-harming behaviour [36]. 

#### 3.5.2. Barrier 2: Young Adults Own Fears about Consequences Inhibits Disclosure

Young adults reported that their own fears, mainly about the consequences of disclosure, limited their honesty in GP appointments. Across the studies, a wide variety of fears regarding consequences were raised by young adults. These included: fear that they would be “considered ‘crazy’” [35] or a “mental health patient” [36]; concerns about confidentiality and who would be told about their suicidal behaviour and self-harm, and what would be recorded in their medical files [36,38,40]; fear that they would be admitted to hospital [38]. 

#### 3.5.3. Barrier 3: Young Adults Are Not Confident GPs Have the Skills to Manage Conversations about Suicide

Many young adults viewed GPs as not being capable of handling young adults presenting with thoughts of suicide or mental health issues and this negatively affected full disclosure. GPs’ lack of technical knowledge regarding suicide and mental health was identified. [36,38,40]. This included “GPs focusing on physical health and not fully investigating suicide risk or mental health issues with young adults” [40]. However, most skill deficiencies identified by young adults could be viewed as GPs lacking therapeutic skills. For example, young adults reported that: “GPs need wider training that taught them how to ‘just be around someone’ who was experiencing mental health difficulties” [36]; GPs do not know how to communicate with young adults and used terminology they did not engage with [40]; GPs do not fully explore their problems and young adults feel that this lack of completeness compromised recommended treatments [40] and resulted in an over-reliance on prescribing medication as a management option [38]. 

#### 3.5.4. Barrier 4: Young Adults Want GPs to Initiate the Conversation about Suicide 

Young adults reported that GPs relying on young adults to raise thoughts of suicide was a barrier to disclosure. They cited a number of reasons for this. Young adults are not all literate in mental health and, consequently, they may not realise they are unwell and so do not disclose thoughts of suicide to GPs [40]. Further, they may not know what support a GP could offer, so once again, do not disclose [40]. Young adults expressed the “belief that it is the responsibility and part of the role of the GP to ensure they assess for mental health difficulties, particularly if there is a prior history or evident risk factors” [36]. Moreover, young adults endorsed the view that having thoughts of suicide, and experiencing common accompanying emotions, such as shame, hopelessness, and burdensomeness, were a barrier for young adults in being able to disclose thoughts of suicide to the GP [36]. Thus, they relied upon the GP to initiate the conversation on difficult personal issues [42] and to pick up any hints they may be dropping about how they were feeling [36]. 

#### 3.5.5. Barrier 5: Time-Limited Consultations Inhibit GPs’ Ability to Identify STB

Young adults recognised that short consultation times made it difficult for GPs to explore and discuss suicide. They could see that the doctors felt rushed [35]. Young adults said short consultation times were “not sufficiently long enough for them to be able to disclose and talk about their mental health difficulties” [36], and that it “impacted the GPs ability to identify problems, see the whole picture, and hindered the development of a genuine connection with the young person” [40].

#### 3.5.6. Facilitator 1: Good Therapeutic Skills Are Essential to Aid Disclosure

GPs having good therapeutic skills were seen as essential if GPs were to assist young adults to safely disclose thoughts of suicide. Young adults reported that they “did not mind being asked about self-harm as long as it was in an empathetic way” [35]; that they “wanted their GPs to be friendly” and that “attentive body language, including eye contact and posture, and demonstrating active listening were important to young adults when [they were] communicating about suicidal behaviours or self-harm” [40]. Young adults felt that it was important that the GP adapt to how the young person is presenting and use this information to guide their approach, for example, being more direct in their questioning when the young person is having difficulty finding the words to explain how they are feeling [36]. They appreciated GPs who were “welcoming, attentive and made an effort to build rapport” [36]. Good therapeutic skills helped GPs create a safe space in which young adults could talk openly about their mental health difficulties including STB [36]. 

#### 3.5.7. Facilitator 2: Extra Support Assists Young Adults to Disclose and Discuss STB 

Young adults reported that extra time in appointments, such as double appointments, allowed them more space to talk and were helpful in facilitating disclosure [35]. Providing follow-up appointments was also seen as helpful [36,40]. Most young adults found it easier to talk about and disclose information relating to their suicidal experience after getting to know their GP [36]. Young adults also wanted GPs to explain any resources provided and to spend time with them to rehearse the use of these resources, such as how to call helplines [40]. For some having another person such as a parent attend the appointment was described as beneficial to facilitating disclosure. Young adults felt others could articulate on their behalf and some reported they felt they would be taken more seriously by the GP if they had another person with them [36]. 

#### 3.5.8. Facilitator 3: Young Adults “Want a Collaborative Dialogue with GPs” [40] 

Young adults reported a desire for a collaborative dialogue with GPs that allowed them autonomy and control within the appointment and over any treatment suggested in the appointment [40]. Young adults indicated they wanted GPs to begin with “proactively exploring their suicidal behaviours and self-harm as part of a collaborative dialogue” [40] and extending this to treatment being offered; for example, stepping out the range of interventions available to address concerns, with medication as a last resort [35]; “being given autonomy to make informed choices about treatment” including the reasons for and consequences of treatment options [40]; “being kept informed by GPs about the outcomes of sharing any information” with others, for example, other services or family members [40]. 

## 4. Discussion

The aim of this review was to understand the barriers and facilitators facing GPs and young adults in raising and discussing suicide in general medical appointments. The current review demonstrated that GPs find it difficult to identify and manage suicide risk in patients across all age groups not just with young adults; and the lack of training and access to specialist services for patients hinders their ability to competently manage those who are suicidal within primary care [18,45,46]. Young adults indicated that they did not understand the role GPs played in mental health [28,47,48] and that even if they did disclose concerns, they believed that GPs lacked the necessary skills to respond to their needs [47,48,49]. They also said that their mental health illiteracy and fears about the consequences of disclosure deterred them from discussing their mental health concerns, including suicidal thoughts, with GPs [50,51,52]. In concurrence with GPs, young adults felt that there was insufficient time in medical appointments to raise mental health concerns including suicidal thoughts [28,49].

Yet, this review did highlight some unique insights from both GPs and young adults in regard to raising and discussing suicide within a medical appointment. It is evident from this review that not all GPs have the confidence in their technical and therapeutic skills to assess and manage suicidal young adults. For some, this may be reflected in their expressed negative attitudes towards working with young adults. This, combined with the limited time available within a typical GP appointment, means that GPs may subconsciously avoid raising suicide with young adults. It is also apparent from the comments made by young adults that they are aware GPs are not necessarily competent to deal with their issues and this makes them reluctant to disclose suicidal thoughts. Unique to this study is the finding that young adults want GPs to initiate the conversation about suicide, rather than rely upon them to voluntarily disclose. Young adults viewed this as the responsibility of GPs who have the knowledge and training and that their own fears and lack of mental health literacy hindered disclosure. Overall, these findings could provide explanations for the failure of GPs to identify mental health issues in young adults [11,15]. These same reasons could be applicable to the failure of GPs to identify suicide risk in young adults. Future studies could explore this dynamic in more detail.

### 4.1. Strengths and Limitations

This study’s strength is that it explores in detail an area not previously examined. The combined perspectives of GPs and young people provide unique and insightful findings. These findings could provide stimulus for new ways to address this important issue.

However, some of the conclusions drawn may be limited by the following. First, there was a scarcity of papers on this topic. With the exception of one paper, from Nicaragua [30], all the papers were from Australia and the United Kingdom (UK) [35,36,37,38,39,40,41,42]. This may limit the relevance of the findings to other countries. However, this review did identify more similarities than differences across the studies despite the disparity in wealth and resources. For example, GPs in all studies endorsed mostly the same barriers and facilitators to raising and discussing suicide in medical appointments. The only difference between Nicaragua, and Australia and the UK was that GPs in Nicaragua explicitly requested additional human and physical resources, such as more space or facilities to see young people privately when discussing sensitive concerns such as suicide. This was consistent with a country whose primary care system has substantially less investment compared to the UK and Australia [30,44]. Otherwise, GPs from Nicaragua agreed with GPs from Australia and the UK that training and more time in consultations would facilitate them to better explore suicide risk with young adults. Further, both GPs in Australia and Nicaragua said that better integrated public health services would also assist them to improve their support to young adults who are suicidal. Overall, this suggests evidence for the generalisability of the GP findings beyond the three countries included in this review.

Second, some of the conclusions drawn from the GP articles may have included some bias. Two of the five GP studies in this review included researcher exploration of proposed interventions to aid GPs. One study explored the utility of self-help materials to support GP consultations with young people on self-harm [35] and the another explored items for inclusion in specialist training for GPs on youth suicide [37]. Due to the researchers’ focus on the development of these interventions, there may have been an over-emphasis on the need for training and resources as a facilitator to enable GPs to overcome the barriers GPs face in raising and discussing suicide in medical appointments. Another two articles included practice nurses who participated along with the GPs in the interviews and focus groups exploring barriers and facilitators to raising and discussing suicide with young adults [30,35]. Research suggests that some nurses hold negative attitudes towards working with suicidal patients [53], but it is unclear how much this may vary when compared to GPs’ attitudes [54]. The data extraction for the thematic analysis did not include any direct quotes from nurses, but the data from these papers may have been influenced by the nurses’ inclusion in the interviews and this, in turn, may have influenced the barriers and facilitators identified in this review. 

Third, all of the papers included in this review were completed prior to the COVID-19 pandemic. The impact of the COVID-19 pandemic on young people and suicide is still evolving [55,56]. As no studies on this topic have been published from this period, further research will be needed to confirm whether the themes identified in this review have been altered as a result of the COVID-19 pandemic.

### 4.2. Implications for Research and Practice

Strong primary care contributes to a well-functioning healthcare system [57] and GPs are pivotal to this system. In this review, GPs reported that they needed extra support and resources if they are to better manage patients who are suicidal. In particular, they identified simpler and clearer referral processes to specialist services and access to in-house multi-disciplinary teams. Some of these interventions have been investigated in the past, mainly in relation to supporting GPs in managing mental illness in primary care. The outcomes have been mixed. For example, improving the referral process to specialist services has not necessarily assisted GPs and, in some cases, added more work [58]. However, tailored training for GPs outlining referral criteria for local specialist services did improve relationships between GPs and specialist services and facilitated some referrals for GPs [59]. On-site mental health workers and psychiatric liaison services to support GPs have been found to be beneficial for GPs and patients in the management of mental health issues [60,61]; however, it is unclear how implementing these initiatives would directly assist GPs in managing suicidal patients. More recently, programs have been designed to directly target suicidal patients. For example, the Hospital Outreach Post-suicidal Engagement (HOPE) provides enhanced support and assertive outreach for people who have attended hospital after treatment for a suicide attempt or are expressing serious planning or intent [62]. Currently, patients are referred to this program only by hospital staff; however, providing GPs with referral access to such programs for suicidal patients, may be more beneficial and cost-effective than providing them with the support they requested in this review. 

Many interventions that may support GPs, such as implementation of the HOPE program, are reliant on primary healthcare funding. However, there are some interventions that are within the remit of GPs to improve. Fifteen percent of young adults (aged 15–24) consult a GP for mental health issues [2]. Therefore, it is critical that GPs receive specialist training on assessing suicide risk in young adults. Training needs to be comprehensive and address the many deficiencies identified in the papers in this review and by other researchers. These include: increasing time spent in the undergraduate curriculum on child and adolescent development so doctors can identify mental illness in young adults, and improving doctors’ knowledge and skills in working with young people. For example, knowledge on how to assess risk in young adults with an explanation of risk factors and warning signs of suicide for young people; communication and therapeutic skills including managing challenging consultations with young adults and/or those with complex presentations; how to manage doctor–patient confidentiality when other parties such as parents or schools/universities are to be involved; how to access specialist services and/or support for young adults who are at risk of suicide [37,41,63,64]. Sufficient comprehensive training will increase GP-perceived competency [65] and will, in turn, decrease the anxiety and discomfort experienced by some GPs when working with suicidal patients [66]. GPs who perceive themselves as competent to work with suicidal patients are more likely to enquire about suicide [18,19], and are more willing to assess and treat suicidal patients [29,65]. Assessing the effectiveness of training will also be important. Self-efficacy measures have been found to be more effective at assessing GP competencies compared to using the completion or recency of training [66]. This type of measure could be administered prior to and post-completion of training and/or after a period of utilising the training in practice. General Practitioners are time-poor, so it will be a challenge to improve training for all, particularly for those that are already in practice. It could be that the undergraduate curriculum should be the priority for the creation of training content. 

However, in the short-term, GPs will continue to face time constraints and heavy workloads. Thus despite best intentions and training, GPs may not always actively explore suicide risk unless the young adult raises concerns. Screening is one tool that could be implemented to assist GPs to identify those patients who would most benefit from further assessment. Suicide risk screening for high-risk populations, such as young people, has been advocated as a means of proactively identifying and triaging those who may need treatment and support [67]. Increasingly, suicide risk screening is being used in the USA across multiple settings including paediatric hospitals and college health services with the aim of identifying patients who have thoughts of suicide but are presenting with non-psychiatric complaints [68,69]. Most screening is either administered in person or via a paper-based screen prior to the appointment. However, both could be time-consuming for GPs to implement. Recent studies have found that young adults are more open to answering sensitive questions on topics that they find difficult to raise directly with a GP, such as self-harm, in online questionnaires completed prior to a GP appointment [70]. While concerns exist that screening tools, particularly short screens, are not accurate in predicting suicide [25,71], their utility is in alerting GPs to the possibility of suicide risk. GPs can then focus on initiating an assessment with those most at risk and exploring the supports a young person may need. Patients report these types of online screening tools as acceptable [69,70]. Furthermore, young adults appreciate the opportunity to flag concerns prior to appointments and report greater satisfaction with GP appointments after the use of such tools [70]. The challenge once again for GPs will be introducing such tools into general practice alongside all their other competing demands.

## 5. Conclusions

Both GPs and young adults agreed that GPs needed to improve their skills if young people are to be comfortable to raise and discuss their thoughts of suicide and suicidal behaviours in medical appointments. Training is one initiative that should be pursued by GPs. This training needs to be comprehensive and focused on improving GPs’ competency in working with young adults. GPs have made other suggestions on facilitators that could assist them to improve their ability to identify and manage young adults who are suicidal in primary care. However, many of these initiatives may be beyond the capacity of GPs to implement, requiring those who fund the primary care systems to develop, fund, and implement these initiatives for GPs. It is important to implement interventions for GPs that will help to overcome young adults’ reluctance to voluntarily disclose thoughts of suicide. However, this review has highlighted the need for GPs to proactively assess young adults for suicide risk rather than relying on them to voluntarily disclose concerns. Introducing suicide risk screening in primary care for young adults is one intervention GPs could initiate independently. This should be a priority intervention for GPs to implement so they can identify and help young people who are experiencing thoughts of suicide.

## Figures and Tables

**Figure 1 ijerph-20-00822-f001:**
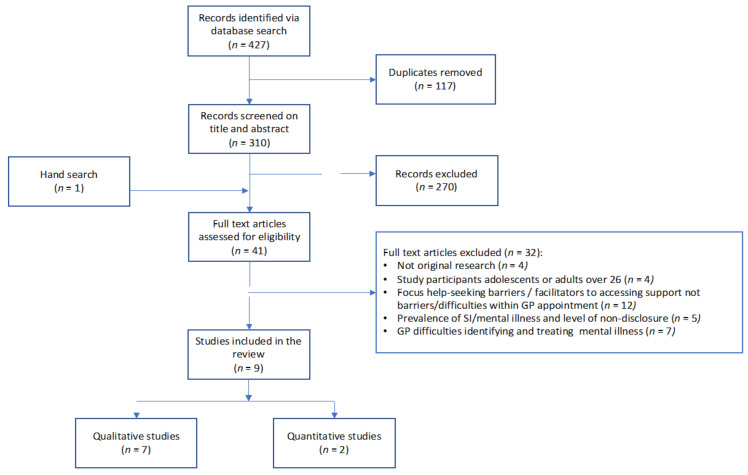
Flow diagram.

**Table 1 ijerph-20-00822-t001:** Study characteristics.

#	Study	Location	Participants	Age	% Female	Sample Type	Selection	Type	Data Collection Approach
*Qualitative Studies*									
1	Bailey et al. 2019 [35]	UK	14 GP 16 PN 15 YA	n.r. n.r.	36% 47%	GP practices (near universities) YA with previous self-harm / GP attendance	Purposively Selected Self-selected	Qual.	Focus groups
2	Bellairs-Walsh et al. 2020 [40]	Australia	10 YA	16–24 *M* = 20.7	70%	YA with experience of discussing self-harm with GP	Self-selected	Qual.	Focus groups
3	Bellairs-Walsh et al. 2021 [41]	Australia	15 GPs	37–53 *M* = 45	40%	GP practices (youth targeted)	Purposively selected	Qual.	Individual interviews (5) Group interviews (3)
4	Farr et al. 2021 [36]	UK	8 YA	17–23	75%	YA with history of suicide attempts / registered with GP	Self-selected	Qual.	Semi-structured interviews
5	Michail and Tait 2016 [37]	UK	28 GP	*M* = 37	68%	GP practices in one area of UK	Self-selected	Qual.	Focus groups (4); semi-structured interview (1)
6	Mughal et al. 2021 [38]	UK	13 YA	19–25	92%	YA with previous self-harm	Self-selected	Qual.	Semi-structured interviews: face-to-face (9), telephone (4)
7	Obando Medina et al. 2014 [30]	Nicaragua	7 GP 5 PN	n.r. n.r.	n.r. n.r.	GP primary healthcare centres. Contact with young adult in daily work	Purposively selected	Qual.	Semi-structured interviews
*Quantitative Studies*									
8	Beckinsale et al. 2001 [42]	Australia	364 YA	15–24	65.9%	YA with thoughts of suicide attending GP	Self-selected convenience sample	Quan.	Practice audit / Cross-sectional survey
9	Michail et al. 2017 [39]	UK	70 GP	*M* = 47	57%	GP practices in one area of UK	Self-selected	Quan.	Cross-sectional survey

YA—Young adults; n.r. Not reported; UK = United Kingdom; PN = Practice nurse; *M* = Mean; Qual. = Qualitative; Quan. = Quantitative

**Table 2 ijerph-20-00822-t002:** GPs: Key barrier and facilitator themes (n= 5).

Theme	#	Barrier Theme	Sub-Theme	*n*	Studies
Barriers	1	GPs attitudes’ impede enquiry about suicide with young adults		5	[30,35,37,39,41]
	2	GPs find appointments with young adults uniquely difficult	Communication is difficult	4	[30,35,39,41]
			Distinguishing suicidal behaviour is difficult	3	[30,37,41]
			Managing confidentiality is difficult	3	[30,37,41]
	3	GPs have insufficient time and asking about suicide adds stress to their day		4	[30,35,37,41]
	4	GPs feel ill-equipped to assess and manage suicidal young adults		3	[30,37,41]
Facilitators	1	Training and resources to improve GP ability to detect and assess STB		4	[30,35,37,41]
	2	More time for consultations and regular Reviews		3	[30,37,41]
	3	Improving GP therapeutic skills		3	[35,37,41]
	4	Better systems to support GP ability to manage suicidal young adults in primary care		2	[30,41]

STB: Suicide thoughts and behaviours.

**Table 3 ijerph-20-00822-t003:** Young adults: Key barrier and facilitator themes (*n* = 5).

Theme	#	Barrier Theme	*n*	Studies
Barriers	1	GP responses (verbal and non-verbal) affect decisions whether to disclose STB	4	[35,36,38,40]
	2	Young adults’ own fears about consequences inhibit disclosure	4	[35,36,38,40]
	3	Young adults are not confident GPs have the skills to manage conversations about suicide	3	[36,38,40]
	4	Young adults want GPs to initiate the conversation about suicide	3	[36,40,42]
	5	Time-limited consultations inhibit GPs ability to identify STB	3	[35,36,40]
Facilitators	1	Good therapeutic skills are essential to aid disclosure	3	[35,36,40]
	2	Extra support assists young adults to disclose and discuss STB	3	[35,36,40]
	3	Young adults want a collaborative dialogue with GPs	2	[35,40]

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
