# Peer review of "Raising Suicide in Medical Appointments—Barriers and Facilitators Experienced by Young Adults and GPs: A Mixed-Methods Systematic Review"

_ijerph, 2023, doi:10.3390/ijerph20010822_

Round 1

Reviewer 1 Report

Thank you for the opportunity to review this paper. It provides new insights into the possible reasons behind why clients and GP's do not address the issue of suicide in GP appointments. This in turn provides food for thought as to how one might address this issue in practice and how to manage it better for both GP's and clients. 

Author Response

Thanks for your very positive feedback on our article.

Reviewer 2 Report

The work presented is of enormous interest and relevance. It is worth paying attention to the barriers and facilitators to addressing suicide. Even more so when suicide has become a priority issue in international public health. 

The proposal of the paper is also very relevant for clinicians as it considers key aspects in the intervention of general practitioners.

It would be interesting if the authors could expand on the strengths and, especially, the limitations of the study. This is fundamental, since, among other issues, the authors start with a very broad objective that includes both patients and professionals.

It is also essential to expand on the implications for practice. This can be very relevant for clinicians.

The conclusions section is very brief. It is worthwhile to go deeper into this.

Author Response

Thank-you for your feedback. Please see the attachment for our responses.

Reviewer 3 Report

I enjoyed reading this systematic review. I particularly like the thematic analysis.

However, upon reflection, I question a couple of points:

-The article provides a great review period, but I question the inclusion of some of the articles. More specifically, the article from Nicaragua does not fit the initial framing. Also, in terms of timeline, I question the inclusion of an article from 2001. If you plan to include these articles, I would advise some justification. Overall, in terms of timeline, the authors do not directly address the impact of the COVID pandemic on suicidal ideation, planning, and attempts. 

-I also question the use of the terminology STBs, suicidal ideation, planning, and attempts is more frequently used as it arguably provides a greater level of nuance.

-In the introduction, do the authors mean rates of completed suicides?

-The majority of cases are from the UK and Australia. I assume that there are specific protocols for GPs? (i.e., a referral process to mental health services etc). GPs are important, but in the discussion, I think the role the GP plays in the system could provide greater accuracy.

-In the discussion, literature from France is referenced. I would question if these comparisons can be made (i.e., between France and the UK/AUS)

-Line 412 stops short. I think more is needed here.

-Line 420 italics?

-There are spacing issues throughout the manuscript after the full stop. Sometimes one space, sometimes two spaces.

-Line 48 needs a reference to support.

I hope the authors find these comments useful. Best wishes.

Author Response

Thank-you for your feedback. Please see the attachment for our responses to your feedback.

Round 2

Reviewer 2 Report

The modifications made by the authors comply with the requirements made by this reviewer.